# StreetAware: A High-Resolution Synchronized Multimodal Urban Scene Dataset

**DOI:** 10.3390/s23073710

**Published:** 2023-04-03

**Authors:** Yurii Piadyk, Joao Rulff, Ethan Brewer, Maryam Hosseini, Kaan Ozbay, Murugan Sankaradas, Srimat Chakradhar, Claudio Silva

**Affiliations:** 1Tandon School of Engineering, New York University, Brooklyn, NY 11201, USA; 2NEC Laboratories America, Inc., Princeton, NJ 08540, USA

**Keywords:** urban sensing, urban multimedia data, urban intelligence, street-level imagery, data synchronization, computer vision

## Abstract

Access to high-quality data is an important barrier in the digital analysis of urban settings, including applications within computer vision and urban design. Diverse forms of data collected from sensors in areas of high activity in the urban environment, particularly at street intersections, are valuable resources for researchers interpreting the dynamics between vehicles, pedestrians, and the built environment. In this paper, we present a high-resolution audio, video, and LiDAR dataset of three urban intersections in Brooklyn, New York, totaling almost 8 unique hours. The data were collected with custom Reconfigurable Environmental Intelligence Platform (REIP) sensors that were designed with the ability to accurately synchronize multiple video and audio inputs. The resulting data are novel in that they are inclusively multimodal, multi-angular, high-resolution, and synchronized. We demonstrate four ways the data could be utilized — (1) to discover and locate occluded objects using multiple sensors and modalities, (2) to associate audio events with their respective visual representations using both video and audio modes, (3) to track the amount of each type of object in a scene over time, and (4) to measure pedestrian speed using multiple synchronized camera views. In addition to these use cases, our data are available for other researchers to carry out analyses related to applying machine learning to understanding the urban environment (in which existing datasets may be inadequate), such as pedestrian-vehicle interaction modeling and pedestrian attribute recognition. Such analyses can help inform decisions made in the context of urban sensing and smart cities, including accessibility-aware urban design and Vision Zero initiatives.

## 1. Introduction

Driven by continuous improvements in computational resources, bandwidth optimization, and latency, activity-rich traffic intersections have been implicated as excellent locations for smart city intelligence nodes [1]. Audio and video sensors located at intersections are, thus, capable of generating large amounts of data. Concomitantly, deep learning and edge computation of these data allow, in real-time, the geospatial mapping and analysis of urban intersection environments, including moving entities, such as pedestrians and vehicles. Intersections are some of the most critical areas for both drivers and pedestrians. They are where vehicles and pedestrian paths most frequently cross. Globally, pedestrians represent 23% of the 1.35 million worldwide road traffic deaths every year with most events occurring at pedestrian crossings [2,3]. Thus, predicting pedestrian trajectories at intersections and communicating this information to drivers or assisted/autonomous vehicles could help mitigate such accidents. Understanding an intersection scene has significant implications for self-driving vehicles in particular. Figure 1 outlines the concept of enhancing the safety of traffic participants by providing real-time insights into out-of-sight events at intersections using a combination of multimodal sensing and edge and in-vehicle computing. In this example, pedestrians and (semi)autonomous cars are sensed by sight (cameras) and sound (microphones) at intersections, and information is relayed to each car’s self-driving system. In this process, edge computing and via the cloud helps extract, in real-time, useful information from the data.

Within the navigation system of an autonomous vehicle, it is important that its control system has detailed, accurate, and reliable information as it approaches such a scene to determine, for instance, the number of road entries into an upcoming crossing or pedestrian and vehicle trajectories to avoid collisions [4]. For such purposes, urban analytical data should have high precision, granularity, and variation (such as multiple perspectives of the same area) to be effectively useful.

In this study, we present 7$\frac{3}{4}$ h of synchronized data collected at urban intersections by specialized Reconfigurable Environmental Intelligence Platform (REIP) sensors developed by the Visualization and Data Analytics (VIDA) Research Center at NYU [5]. REIP sensors are capable of dual 5 MP video recording at 15 fps as well as the 12-channel audio at 48 kHz for the recording of pedestrian and vehicle traffic at various locations. We selected three intersections in Brooklyn, New York, with diverse demographic, urban fabric, and built environment profiles and equipped each with four REIP sensors. The sensors were placed at each corner of the intersection and recorded the dynamic of pedestrian and vehicle interaction for several ≈40 min sessions, resulting in a total of ≈2 TB of raw audiovisual data. The data were synchronized across all sensors with high accuracy for both modalities (one video frame and one audio sample, respectively) using a custom time synchronization solution detailed later. High-synchronization is important so that events that happen across cameras and between video and audio can be viewed and analyzed together with reduced effort, and with confidence, those events actually occurred at the time inscribed in the data.

The presented dataset, which we call StreetAware, is unique to other street-level datasets such as Google Street View because of the following combination of characteristics:Multimodal: video, audio, LiDAR;Multi-angular: four perspectives;High-resolution video: 2592 × 1944 pixels;Synchronization across videos and audio streams;Fully anonymized: human faces blurred.

To demonstrate these key features of the data, we present four uses for the data that are not possible on many existing datasets — (1) to track objects using the multiple perspectives of multiple cameras from both audio (sound-based localization) and visual modes, (2) to associate audio events with their respective visual representations using audio and video, (3) to track the amount of each type of object in a scene over time, i.e., occupancy, and (4) to measure the speed of a pedestrian while crossing a street using multiple synchronized views and the high-resolution capability of the cameras.

Our contributions include:The StreetAware dataset, which contains multiple data modalities and multiple synchronized high-resolution video viewpoints in a single dataset;A new method to synchronize high-sample rate audio streams;A demonstration of use cases that would not be possible without the combination of features contained in the dataset;A description of real-world implementation and limitations of REIP sensors.

The data presented here will allow other researchers to carry out unique applications of machine learning to urban street-level data, such as pedestrian–vehicle interaction modeling and pedestrian attribute recognition. Such analysis can subsequently help inform policy and design decisions made in the context of urban sensing and smart cities, including accessibility-aware design and Vision Zero initiatives. Among the other possibilities we discuss later, further analysis of our data can also shed light on the optimal configuration needed to record and analyze street-level urban data.

Our paper is structured as follows. In Section 2, we review some of the literature on street-view datasets and how these types of data have been analyzed with deep learning. In Section 3, we discuss our custom sensors and detail data acquisition and processing, with emphasis on the precise synchronization of multiple data modalities (i.e., audio and video). We lay out the motivation for and demonstrate the potential applications of the data in Section 4 and provide a discussion and concluding remarks in Section 5 and Section 6 correspondingly.

## 2. Related Work

In this section, we will review some of the currently available audiovisual urban street-level datasets, then succinctly review applications of such data related to deep learning-based object detection and pedestrian tracking and safety.

### 2.1. Datasets

A handful of related datasets exist. The first is the popular Google Street View [6]. Released in 2007, at a time when a limited number of cities had their own street-level photography programs, Google Street View was revolutionary in that it combined street-level photography with navigation technology. Publicly available but not entirely free, Google Maps Street View includes an API and extensive street-level image coverage throughout much of the World’s roadways. Unlike StreetAware, Google Street View is a collection of disparate images instead of stationary video recordings of specific places. Moreover, Google Street View often has multiple viewpoints that are in close proximity to one another, but they do not overlap in time. Therefore, synchronization across multiple views is not possible. Another dataset is Mapillary [7]. Mapillary street-level sequences contain more than 1.6 million vehicle-mounted camera images from 30 major cities across six continents, distinct cameras, and different viewpoints and capture times, spanning all seasons over a nine-year period. All images are geolocated with GPS and compass, and feature high-level attributes such as road type. Again, these data are not video or synchronized and do not include audio. The next dataset is Urban Mosaic [8], which is a tool for exploring the urban environment through a spatially and temporally dense data set of 7.7 million street-level images of New York City captured over the period of one year. Similarly, these data are image-only and unsynchronized across views. Another street-level urban data set is SONYC [9]. SONYC consists of 150 million audio recording samples from the “Sounds of New York City” (SONYC) acoustic sensor network and is aimed at the development and evaluation of machine listening systems for spatiotemporal urban noise monitoring. However, SONYC does not contain visual information. Finally, there is Urban Sound & Sight (Urbansas) [10], which consists of 12 h of unlabeled audio and video data along with 3 h of manually annotated data, but does not contain multiple views. These and other street-level datasets (most oriented toward self-driving vehicle research) are listed in Table 1 with brief descriptions of each. StreetAware is unique in that it combines stationary, multi-perspective, high-resolution video and audio in a synchronized fashion.

### 2.2. Deep Learning Applications

A number of recent studies have explored the use of deep learning for detecting and analyzing objects in street-level audio and video data. A study by Zhang et al. [23] developed an approach to automatically detect road objects and place them in their correct geolocations from street-level images, relying on two convolutional neural networks to segment and classify. Doiron et al. [24] showed the potential for computer vision and street-level imagery to help researchers study patterns of active transportation and other health-related behaviors and exposures. Using 1.15 million Google Street View (GSV) images in seven Canadian cities, the authors applied PSPnet [25], and YOLOv3 [26] to extract data on people, bicycles, buildings, sidewalks, open sky, and vegetation to create associations between urban features and walk-to-work rates. Charitidis et al. released a paper in 2023 [27] in which they utilized several state-of-the-art computer vision approaches, including Cascade R-CNN [28] and RetinaFace [29] architectures for object detection, the ByteTrack method [30] for object tracking, DNET architecture [31] for depth estimation, and DeepLabv3+ architecture [32] for semantic segmentation to detect and geotag urban features from visual data. Object detection systems have also been specifically developed for the collection and analysis of street-level imagery in real-time [33]. In “Smart City Intersections: Intelligence Nodes for Future Metropolises” [1], Kostec et al. detail intersections as intelligence nodes using high-bandwidth, low-latency services for monitoring pedestrians and cloud-connected vehicles in real-time. Other computer vision applications to urban street view imagery include extracting visual features to create soundscape maps [34], mapping trees along urban street networks [35], estimating pedestrian density [36] and volume [37], associating sounds with their respective objects in video [10], and geolocating objects from a combination of street-level and overhead imagery [38].

Pedestrian speed and trajectory prediction are some of the primary computer vision goals in the urban data analytical community, especially in the field of advanced driver assistance systems [3]. The performance of state-of-the-art pedestrian behavior modeling benefits from recent advancements in sensors and the growing availability of large amounts of data (e.g., StreetAware) [39]. A study by Kuo et al. [40] compared estimations of pedestrian speed from a classical model and a neural network in corridor and bottleneck experiments, with results showing that the neural network can better differentiate the two geometries and more accurately estimate pedestrian speed. Ahmed et al. [41] sought to use a fast region-convolutional neural network (Fast R-CNN) [42], a Faster R-CNN [43], and a Single Shot Detector (SSD) [44] for pedestrian and cyclist detection based on the idea that automated tracking, motion modeling, and pose estimation of pedestrians can allow for a successful and accurate method of intent estimation for autonomous vehicles. Other related studies in the literature include applying deep learning techniques for the prediction of pedestrian behavior on crossings with countdown signal timers [45], mapping road safety from street view imagery using an R-CNN [46], and identifying hazard scenarios of non-motorized transportation users through deep learning and street view images in Nanjing, China [47].

## 3. The StreetAware Dataset

In this section, first, we will review existing audiovisual sensor options and make the case for harnessing the REIP sensors (Figure 2) designed and constructed in our lab. Next, we describe how the data are collected, processed, and synchronized.

### 3.1. REIP Sensors

A multi-view requirement of our data collection could easily be satisfied with off-the-shelf video surveillance systems that often include a set of wireless IP cameras. These cameras transmit their video feeds to a central data storage location (in the form of a local hard drive) which can sometimes be synchronized with a cloud but is not required for the system’s operation. The cameras also include a night mode which can prove beneficial during low-light conditions. However, these cameras rarely provide audio because of privacy concerns and rely on manually configured timing information or NTP (network time protocol) for time-stamping of the video. The latter is a significant barrier to a multi-view analysis of fast-moving objects such as cars. A car traveling at 40 mph covers more than a meter of ground per frame when recorded at 15 fps. Therefore, frame-accurate video synchronization is also a requirement for our dataset and, unfortunately, cannot be met with off-the-shelf security cameras, which are also often operating at reduced frame rates due to limited storage.

There exist commercial motion tracking systems that use high-speed cameras synchronized by NTP. Although these cameras provide high temporal resolution and accuracy for video, they would be insufficient for the synchronization of audio data (sub-millisecond timing accuracy required). Furthermore, such cameras are typically designed for indoor infrared light imaging, are costly, and rely on an Ethernet interface for synchronization and data transfer which makes them impractical for larger-scale urban deployments.

Another commercial device that provides quality video with audio at a reasonable price is the GoPro camera. However, the GoPro was designed for independent operation so it does not feature quality synchronization across multiple cameras. Moreover, the synchronization across video and audio modalities is also known to be a problem because of audio lag offset and differences in sampling frequencies. Recently, GPS-based time-coding has been introduced in the latest versions of GoPro cameras. This could help with synchronizing the start of the recordings but does not solve the ultimate problem of long-term synchronization. The time drift caused by manufacturing variations of the internal crystal oscillator’s frequency that drives digital logic (including the sampling frequency) is also susceptible to temperature-based variations. Moreover, there is no way to know when the GoPro is experiencing lost frames during recording, which ruins the single timestamp-based synchronization altogether. The solution would be a continuous (re)synchronization of the cameras from a single clock source during the entire recording process. Other potential issues include remote control and monitoring of the camera’s status as well as weatherproofing that may require external devices and housing depending on the camera version.

Ultimately, the sensors used in this study are custom-built in our lab. An overview of the sensor’s architecture is provided in Figure 2 with its specifications listed in Table 2. The sensor includes two cameras and a microphone array. It also features a high-precision custom synchronization solution for both video and audio data based on a 2.4 GHz radio module receiving common global timestamps from a master radio device. Each camera records 5 MP video at 15 fps and the microphone array records audio through 12 synchronized channels. The custom acoustic front-end was designed to capture audio from the 4 × 3 digital pulse density modulated (PDM) micro-electro-mechanical systems (MEMS) microphones. It uses the USB MCHStreamer as an audio interface which is a USB audio class-compliant device, so it is compatible with the readily available microphone block in the audio library of the REIP SDK [5]. Each sensor has 250 GB of internal storage and is operated on a FlashFish portable power station. The computing core is the NVIDIA Jetson Nano Developer Kit, which offers edge-computing capabilities. The majority of the sensor’s hardware is enclosed within the weatherproof aluminum housing. Heat sinks are designed to offer resistance to extreme temperatures, providing better performance. For sensor control, a locally deployed network router and Wi-Fi connectivity are used.

REIP sensors provide high-resolution video and audio recording with an in-built synchronization solution (the high-level architecture is shown in Figure 2). Both cameras and audio interface are USB 2.0 devices. Of note is the design of the audio pipeline where the MCHStreamer interface is receiving an additional audio-like signal from the microcontroller unit (MCU). The purpose of this signal is to embed the global timing information received by the radio module as additional audio channels. For video, the individual image frames are timestamped by the NVIDIA Jetson Nano as they arrive into the camera block of the data acquisition and processing pipeline powered by REIP SDK. For that, the MCU is also connected to the computing platform via a USB 1.1 interface and continuously provides the latest global timestamp transmitted to each sensor by a master radio module (a separate device).

### 3.2. Data Collection

Three intersection locations were selected to acquire the dataset with different road configurations and pedestrian demographics as described below:*Commodore Barry Park.* This intersection is adjacent to a public school. It has a low-to-medium frequency of traffic making it an uncrowded intersection.*Chase Center*. This intersection is adjacent to the Chase Bank office building within Brooklyn’s MetroTech Center. It is also an active pedestrian intersection.*DUMBO*. The intersection of Old Fulton Street and Front Street is under the Brooklyn Bridge. Being a tourist destination, this intersection is the busiest of the three. Because of smaller crosswalks and heavy traffic, it provides challenges such as occlusion and a diverse range of pedestrian types.

Overhead map locations and the sensors’ positions for the recording sessions at Commodore Barry Park as an example are shown in Figure 3. Each sensor is equipped with two 5 MP USB cameras providing a combined 160${}^{\circ }$ horizontal field of view at a recording rate of 15 fps. The 4 × 3 microphone array of each sensor records at a sampling rate of 48 kHz. Every sensor was powered by a portable power station with a 300 Wh capacity. An Ouster OS-1 LiDAR sensor is also included. It has a configuration of 16 vertical scanning lines at 1${}^{\circ }$ angular resolution and 1024 samples per revolution.

We used four REIP sensors at each intersection, one placed at each corner of the intersection. We recorded several 30–45 min long sessions at each intersection—four at Commodore Barry Park, three at Chase Center, and four at DUMBO. This results in about 200 GB of raw audiovisual data recorded by each sensor per location (limited by the sensor’s max storage capacity of 250 GB). In total, we collected ≈2 TB of raw data.

The data acquisition pipeline of the sensors is shown in Figure 4. The pipeline is based on the software blocks available in the REIP SDK as released in [5]. Because our sensors are based on the budget NVIDIA Jetson Nano computing platform, a slight modification was necessary for a camera block to be able to timestamp every frame from both cameras for synchronization purposes. We bypass the decoding of JPEG images sent by the cameras to free up CPU resources. Instead, we direct the raw video stream to the file using features of GStreamer library [48] that the camera block is based upon. Still, we did experience some lost frames when recording during the summer month of August due to the throttling of the sensor’s NVIDIA Jetson Nano computing platform after prolonged exposure to extreme temperature conditions.

The output of the sensor’s data acquisition pipeline contains three types of data: 500 MB chunks of video data (approximately one minute of recording, depending on the intersection), JSON files containing batches of timestamps for each frame in the video data chunks, and 5-s long chunks of audio data with its timing information embedded as extra audio channels. We spare the users from working with the sensor’s raw data by preprocessing it, including anonymization and synchronization. We also use a space-efficient video codec H.264 instead of the camera’s original MJPEG data stream. Table 3 summarizes the specifications of the processed dataset that we are releasing.

### 3.3. Data Synchronization

In this section, we detail our synchronization techniques, first for audio, then for video data modality. The synchronization techniques are independent for each modality. Figure 5 illustrates the overall principle.

The method is fundamentally reliant on the hardware design of the sensors where the communication delay between the master radio and each sensor’s slave radio is constant, and the radio waves propagation delay is negligible due to the large speed of light of 299,792 km/s. Similarly, the data readout latency for the cameras is equal across sensors because of the identical cameras used. The video modality can then be synchronized with audio by calibrating the frame readout latency. For that, a rapid event with a loud sound, such as a clap, is recorded in close proximity to the sensor (for negligible sound propagation delay). The true time of the event is then deducted based on sound and compared to the latest global timestamp received by the computing platform when the video frame is released by the driver into a REIP pipeline.

#### 3.3.1. Audio

Audio synchronization is a challenging task because audio data are being sampled at a very high rate, 48 kHz in the case of our sensors. Furthermore, the speed of sound wave propagation in the air is c=343 m/s which translates into a synchronization accuracy requirement of less than one millisecond, across all sensors, for any meaningful audio-based sound source locations to work. Such accuracy cannot be achieved by simply attaching a timestamp to the chunks of audio provided by the driver because of the large ’jitter’ of such timestamps caused by the random operating system (OS) interrupts on the computing platform. Therefore, the synchronization information must be embedded into the audio data itself before it even makes it to the audio driver of the OS. In this subsection, we introduce a novel method for high-accuracy audio synchronization by means of embedding a special signal into a dedicated audio channel of the audio interface (Figure 6).

The radio module of each sensor is receiving a global timestamp from a master radio transmitting it at a rate of 1200 Hz. Unlike the operating system of the computing platform, the microcontroller operating the radio module via Serial Peripheral Interface (SPI) can be programmed to process the incoming data packets from the master radio in a very deterministic way. Specifically, the packet arrival interrupt request (IRQ) signal from the radio module causes the MCU to interrupt its current routine and execute a function that decodes the latest timestamp from the data payload of the packet and phase-adjusts the MCU’s internal timer to match the master radio’s clock. The jitter of the continuously adjusted slave clock is less than 1 μs with the nRF24L01+ 2.4 GHz radio module. The timer in turn generates a special synchronization signal connected to one of the inputs of the MCHStreamer device that we use as an audio interface. The MCHStreamer device supports up to 16 channels of synchronous Pulse Density Modulated (PDM) audio recording. An example of how this synchronization signal appears in PCM audio format (converted to by MCHStreamer) is shown in Figure 6.

We are using a simple UART-like serial protocol with one start bit, a 32-bit payload, and a more than 200 audio sample-long stop bit to generate the audio synchronization signal. The start bit and payload bits are five audio samples wide for more reliable encoding. Such a signal is easy to decode during audio processing, and a single audio sample synchronization accuracy is achieved because the start bit of the sequence is aligned with the time of arrival of the timestamp from the master radio, and the micro-controller has a deterministic delay when processing this information. An example of synchronized audio data is shown in Figure 7.

#### 3.3.2. Video

Video recording is inherently occurring at a much lower sampling rate than audio. For instance, the cameras in REIP sensors are configured to record at 15 fps. That corresponds to a ≈67 ms period between consecutive frames. The radio module receives a new global timestamp every ≈0.83 ms, which is almost two orders of magnitude more frequent. Therefore, it makes sense for video recording to timestamp each frame as it is being received by the driver and calibrates the latency between the moment of assignment of this timestamp and when the frame is actually exposed instead of inventing a way of embedding the timing information directly into the image data during exposure as we did for audio. However, this approach comes with new challenges, such as timestamp jitter and lost frames.

There are three timestamps assigned to each video frame: (1) the GStreamer timestamp, which starts from zero and is defined by the camera driver upon arrival of the frame into the queue from the USB, (2) the Python timestamp representing the current system time which is added using the time.time() function when the frame is released by GStreamer into the data acquisition and processing pipeline powered by REIP SDK (Figure 4), and (3) the Global timestamp added to the frame metadata at the same time as the Python timestamp which is the latest global timestamp communicated to the global time block from the MCU via USB 1.1 interface, introducing extra jitter. Figure 8 depicts the jitter progression as it propagates farther down the data acquisition pipeline.

We developed a method for reducing the jitter of global timestamps to virtually zero before rendering the synchronized video streams (Figure 9). The main source of timestamp jitter is operating system interrupts that happen when the computing platform, for example, needs to process various I/O events or perform memory management. That is why GStreamer timestamps have the least amount of jitter because they are defined when the OS handles USB 2.0 data transfers from the camera. That is also why we are starting with GStreamer timestamps to reliably detect if and when there are any frames lost by looking for gaps larger than the expected period of the camera’s frame rate. After correcting for lost frames, we then convert these timestamps into a global timeline through a couple of regressions incorporating the information from other types of timestamps without adding jitter.

In addition to correcting for lost frames and eliminating jitter, our method is also fixing any queue overflow issues that often result in the jamming of multiple frames one after another with very similar Python and Global timestamps. This happens when the queue is emptied out quickly after a prolonged operating system interrupt. Another less common issue is when the frames saved to the disk get corrupted due to high data flow or during the copying of the data from the sensors to a server. The solution requires the corresponding timestamps to be deleted from the metadata, and the associated non-decodable frames are considered lost.

To further validate the video synchronization, we render a surveillance-style mosaic video using processed frames from all eight cameras at a given intersection and a global timeline produced by the synchronization of timestamps. Figure 10 shows a mosaic of the frames at the moment at the Chase Center intersection. Essential for many analysis applications, at any given moment, the recording of all traffic remains in sync from multiple viewpoints. Frames for which a camera did not successfully record data are temporarily made black in the camera’s associated block in the mosaic.

## 4. Use Cases

In this section, we will demonstrate four use cases highlighting the potential applications of StreetAware. First, we present two examples of how such data can enhance pedestrians’ safety in large urban areas by (1) informing pedestrians and incoming traffic of occluded events using multiple sensors and sound-based localization, and (2) associating audio events (such as the presence of loud engines) with their respective visuals. Second, we present easily quantifiable metrics that can be extracted from the data using the StreetAware infrastructure framework: (3) calculating object counts (occupancy) over time, and (4) measuring pedestrian speed during crosswalk traversal.

### 4.1. Audio Source Localization

As the number of sensors deployed in urban environments increases, cities have the potential to become more human-centered by prioritizing pedestrians over cars. Adaptive traffic and pedestrian signal timing is one example of how an intelligent sensing platform can be used to provide a safer environment for pedestrians. By making the signal timing adjustable to the volume of foot traffic as well as the needs of different groups of people, we can allocate longer signal timing to, for example, crowded intersections or pedestrians with special needs such as the elderly, pregnant, or those with vision impairments [45]. Most traffic monitoring systems use one or two fixed cameras for each intersection. However, the complex configuration of intersections in large cities makes it challenging for one or two cameras to count and detect every traffic participant at a busy intersection. They have inherent limitations of fixed field of view and susceptibility to occlusions.

In this first use case, we demonstrate how a synchronized multisensor setting can leverage a data modality, such as audio, to localize sound-emitting traffic participants and reduce the chance an object is completely obstructed by another. The ability of the sensors to “listen” as well as “see” allows the sensor network to remain resilient against occlusions and dead zones. Figure 11 shows an example of detecting the position of a bicyclist using sound, regardless of whether or not the bicyclist is in any of the cameras’ field of view thanks to the diffraction property of the sound waves. In order to reconstruct the position of the bicyclist ringing the bell, we first annotate the high amplitude peaks, $t_{i}$, in the audio data, synchronized using the common time scale as reconstructed from the dedicated audio channel with the serialized timestamps (see Section 3.3.1). With the known sensor positions, $p_{i}$, one can find the sound source position, *p*, at time, *t*, by minimizing the errors:(1)$$p,t=\underset{p,t}{arg min}\sum_{i=1}^{4}{\left(||p-p_{i}||-c\cdot |t-t_{i}|\right)}^{2},$$
where *c* = 343 m/s is the speed of sound in air. All four sensors must hear the sound for this to be a well-posed problem. The results are shown in Figure 11 and are in good agreement with the video footage from the same sensors. There are examples of when audio-based localization was not possible because of noise pollution by a bus and vice versa when the object was out of the field of view of the cameras but could still be heard which illustrates the benefits of such a complementary multimodal approach. This audio-based sound source localization would not be possible without the synchronization technique presented in this paper.

### 4.2. Audiovisual Association

Automatic audiovisual urban traffic understanding is a growing area of research with many potential applications of value to the industry, academia, and the public sector [10]. Deep learning algorithms can leverage video recordings to detect and count a variety of objects in a given scene and calculate specific metrics, such as the distance from one source to another. Although very useful, these algorithms can be improved through augmentation with non-visual data, such as audio. For example, scene understanding can be improved by determining the proximity of out-of-view objects emitting sounds or by detecting loud noises. In addition, local governments may care about noise levels. In New York, for example, city agencies have created laws to automatically monitor and mitigate noise pollution, such as the noise emitted by loud mufflers installed on cars [49]. Thus, sensor networks that include audio, such as the one outlined in this paper, can provide the audio resources necessary to improve urban scene understanding and to monitor city noise.

With StreetAware, in Figure 12, we show how audio can inform the presence of large engines (trucks and buses) at an intersection. Above the video frames, we highlight the corresponding point in time on the acoustic time series (in decibels) extracted from the audio files. With this method, we can easily relate noise peaks to events captured on video. This example shows how StreetAware can advance the state-of-the-art development of audiovisual urban research by providing multiple camera views linked with audio signals to enhance audiovisual recognition algorithms (which are usually trained on single-view video datasets).

### 4.3. Occupancy Tracking & Pedestrian Speed

Stakeholders interested in monitoring the level of activity and quantity of pedestrians and traffic in an area could make use of StreetAware. Here, we present an example in which we evaluate the occupancy of one of the intersections during a recording session. First, the dataset is evaluated with HRNet, an object detection, human pose estimation, and segmentation algorithm. Adapted from the Faster R-CNN network, HRNet is capable of performing state-of-the-art bottom-up segmentation via high-resolution feature pyramids [50]. The network is trained on the COCO dataset [51]. We detect six classes: person, car, bicycle, truck, motorcycle, and bus (Figure 13). Figure 14, in turn, shows example visualizations containing detected objects and human pose outlines. For pose estimation, the model is executed for each “person” detection independently with a focus on that particular bounding box. Such an approach results in temporally consistent pose estimation as the person is walking towards or away from the camera despite significant lens vignetting and brightness variation across the image.

Using this detection framework, Figure 13 shows the total count of the various urban scene entities throughout an entire recording session at the Chase Center intersection. We intentionally choose this particular recording session because it was conducted around 5 p.m. when people are finishing their workday and traveling home. This activity results in a spike in pedestrian and car traffic. There are roughly three times as many pedestrians counted (most crossing a street) toward the end of the recording than at the beginning. This trend also inversely correlates with car count, presumably due to cars yielding the right-of-way to pedestrians. We do not observe as much change in the number of cars or other motorized vehicles because this intersection is typically more consistently busy throughout the day and there is limited space along the street curbs to park cars compared to pedestrians on sidewalks. Parked cars present a certain level of static background count for the car object class.

As outlined in Section 2.2, measuring and predicting pedestrian behavior such as their speed and trajectory are of interest to the computer vision and urban design research communities. Figure 14 presents a simple example of capturing the same pedestrian across two different sensors, highlighting the utility of multiple camera views. The speed of the pedestrian in Figure 14 is manually calculated at 1.1 m/s, derived from traveling ≈11 m (as measured from Google Maps) in 10 s (12,000 global timestamps difference at 1200 Hz update rate). Therefore, one could use a deep learning algorithm and the data’s internal timing to accurately and automatically measure pedestrian speed.

## 5. Discussion

In this study, we collected unique data about traffic and pedestrians from three urban intersections using customized high-resolution audio-video sensors. The novel data includes multiple modalities (audio, video, and LiDAR) with highly accurate temporal information and synchronization. Since the data were recorded in New York City, many demographics are captured. This is particularly important since some of these groups, such as wheelchair users and people with varying levels and types of disabilities, are absent from large-scale datasets in computer vision and robotics, creating a steep barrier to developing accessibility-aware autonomous systems [52]. Identifying pedestrians with disabilities, the qualities of their behavior, and ease at traversing the sensed urban environment is an area of possible exploration with datasets such as this one.

With high-resolution video data, such as in StreetAware dataset, it is important to protect people’s privacy. For that, we leveraged human pose detections to identify where pedestrians are and applied Gaussian blur over the elliptical areas covering their faces. Because automatic methods are not perfect and complete pose detection is particularly susceptible to misdetection in highly crowded areas due to occlusions, we also employ a second model that does direct face detection [53]. Combined with aggressive detection thresholds that result in a high likelihood of producing false positives, we were able to achieve robust video anonymization across the entire dataset.

In Section 4, we demonstrated four uses of the data which are not possible with many other datasets. Section 4.1 provided an example of how the multiple perspectives and audio data can be leveraged for localization of the sound-emitting objects to help overcome visual occlusion by other objects such as large vehicles. Section 4.2 showed that there exist qualitative associations between objects captured by the sensors’ cameras and sounds captured by the sensors’ microphones. More quantitatively, in Section 4.3, we showed that a computer vision model can track the amount and type of objects in our data, confirmed by checking the video and count numbers at specified times. With closer inspection of the occupancy data, one can notice a regular pattern in bus occupancy. Indeed, a bus route does pass through the intersection. This further highlights the importance of our synchronization technique with diligent temporal accounting to correct for any lost frames that might accumulate into a significant time gap in the video footage. Otherwise, attempts at temporal analysis such as, for example, reconstruction of the bus schedule, would suffer from a systematic error.

Finally, in Section 4.3, we presented a simple example of how a coordinated arrangement of multiple synchronized cameras can provide a foundation for pedestrian tracking applications, i.e., unique detection of pedestrians consistent across frames and views. Some currently available software such as NVIDIA’s DeepStream SDK [54] contain built-in C/C++ and Python pipelines for pedestrian tracking. Such tracking technologies could be combined with geo-referenced locations for pedestrians and vehicles to create a map. This digital reconstruction (or “twin”) of an intersection, complete with object locations, can be used for high-level analysis such as determining pedestrian and vehicle counts, travel distances, speeds, and trajectories as they navigate their way through the sensed space.

### Limitations

Overall, the REIP sensors have demonstrated great versatility in data acquisition pipelines and operating conditions. They even withstood, without damage, a sudden rain incident during one of the recording sessions at Commodore Barry Park. The majority of the sensor’s hardware is enclosed within an aluminum weatherproof housing with heat sinks, however, we still experienced occasional periods of lost frames, even during operation in shadows, due to the random operating system interrupts and throttling of the CPUs. Therefore, any long-term deployments would need to account for these issues in a comprehensive way.

The data presented in this study are limited in a few ways. First, the geographic coverage is narrow. Though the activity at each site is somewhat varied, ultimately, data were only collected at three intersections in a single borough in a single highly-developed city in the United States. Second, compared to some other available datasets, StreetAware lacks diverse environmental conditions such as nighttime, precipitation, and fog. However, we did preserve some of the more challenging recording sessions where select sensors experienced an increased amount of occlusion from vegetation during windy conditions. Moreover, third, the data are quite raw—the audio and video recordings are not labeled (e.g., objects, actions, sound sources, etc.) and the LiDAR files provided are unprocessed. In its current form, a user would not be able to query the data for information or have an idea of what is happening over time in a scene without manually inspecting the data or performing further processing and analysis.

## 6. Conclusions

In this paper, we presented the StreetAware dataset, which contains synchronized multi-perspective street-level audio and video in a single dataset. We also presented a new method to synchronize high-sample rate audio streams and demonstrated unique use cases for the data; in the process, we describe the limitations and real-world implementation of REIP sensors.

Moving forward, further applications can be developed to make use of a digital map, such as calculating the distance between vehicles and pedestrians and other vehicles and, thus, the detection of near-collision situations. Aspects unique to StreetAware, such as audio, LiDAR, and multiple in-sync views could be used to augment the performance of such applications (e.g., incorporating the sound of car horns into near-accident detection). Other future areas of investigation include determining the optimal number of cameras to capture the same information captured in this dataset, and the viability of processing the data in real-time on-site (edge computing). Building off the pedestrian detection and speed measurement established here, looking ahead, we intend to evaluate pedestrian and vehicle movement per traffic light cycle. We will leverage the multi-view and synchronization features of the dataset to reconstruct the timing of traffic lights as seen from different camera locations. This will enable us to measure pedestrian and motorist adherence to traffic laws. Other researchers exploring urban street sensing applications that benefit from high-resolution, multimodal, and precisely synchronized data should find this dataset especially useful.

## Figures and Tables

**Figure 1 sensors-23-03710-f001:**
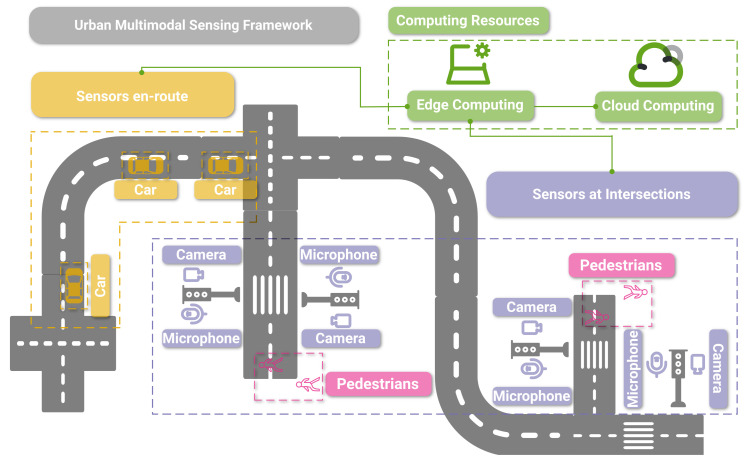
Illustration of the basic concept of combining multimodal sensors at critical nodes (e.g., intersections) with on-device and in-vehicle computing capabilities to provide greater awareness to urban traffic participants.

**Figure 2 sensors-23-03710-f002:**
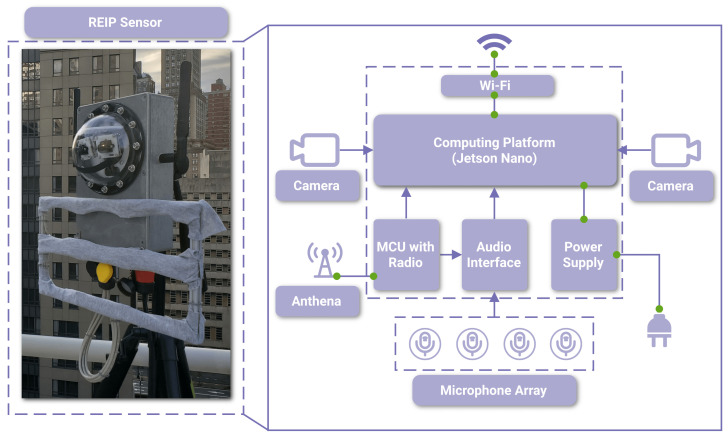
A photo of the REIP sensor in its protective metal housing ready for deployment (**left**) and its internal architecture (**right**).

**Figure 3 sensors-23-03710-f003:**
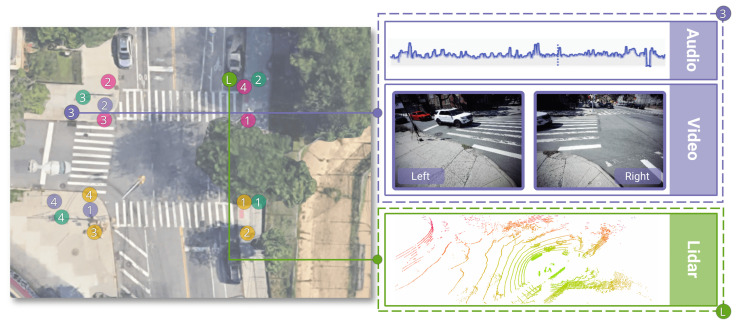
Illustration of the sensor positions and data types at the Commodore Barry Park intersection. Colors denote the different recording sessions, and numbers indicate the REIP sensors. This figure highlights all data modalities that are being captured during the collection process: audio, video, and LiDAR scans. Green L indicates the LiDAR sensor position fixed for all recording sessions.

**Figure 4 sensors-23-03710-f004:**
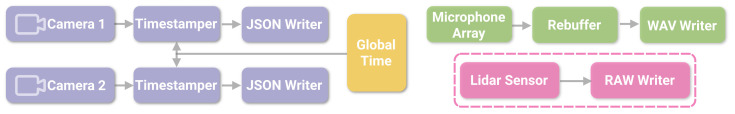
The sensor’s data acquisition pipeline is built using software blocks available in the REIP SDK. It contains separate tasks for each camera and the microphone array. The LiDAR data acquisition is performed on a separate machine (orchestrator laptop).

**Figure 5 sensors-23-03710-f005:**
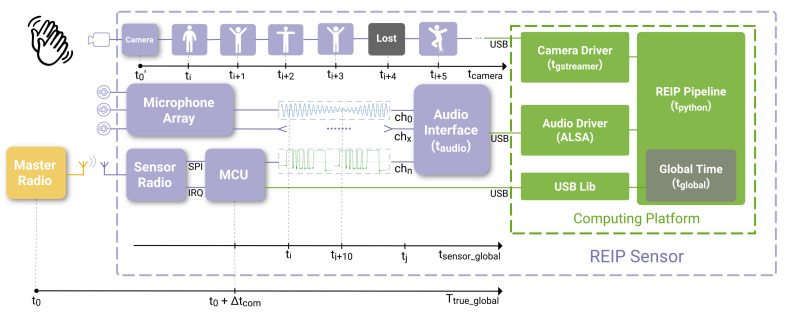
Multimodal synchronization workflow. Each sensor is receiving the global timestamps from a master radio (at 1200 Hz) and is embedding every 10th of them in a serialized form as an extra audio track synchronous with the microphone array data. For the cameras used in REIP sensors, it is not possible to embed the timing information directly into the video data itself. Instead, the timestamps provided by the camera driver are converted into the global timeline using the computing platform that is continuously updating the latest timestamp received by the microcontroller unit (MCU) via USB. More details on video synchronization can be found in Section 3.3.2. The camera’s time axis is compressed by about an order of magnitude for illustration purposes.

**Figure 6 sensors-23-03710-f006:**
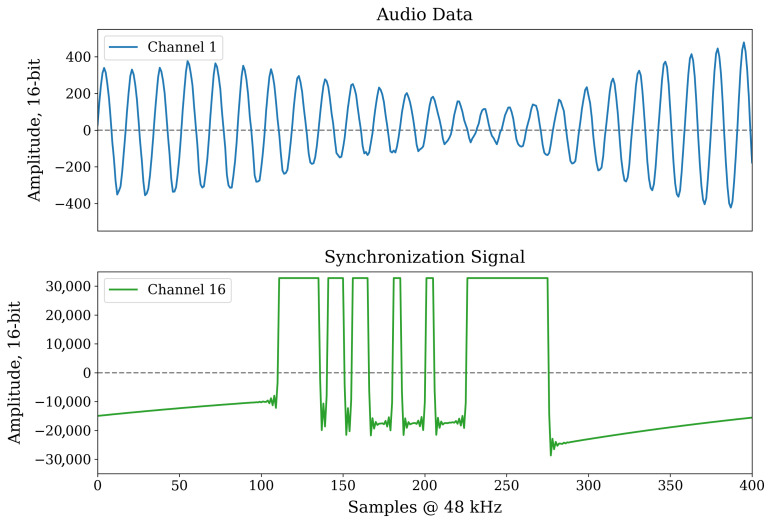
Example of a synchronization signal embedded into the last channel of audio data at a 120 Hz rate. It contains a serialized 32-bit timestamp that is shared across multiple sensors with 1 μs accuracy using a 2.4 GHz radio module. High synchronization accuracy is required due to a high audio sampling rate of 48 kHz.

**Figure 7 sensors-23-03710-f007:**
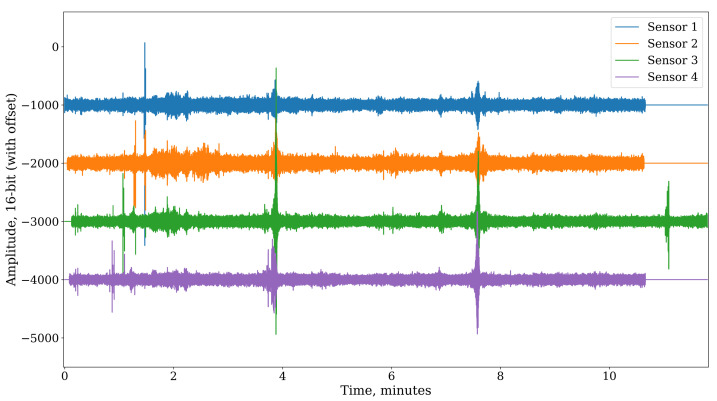
A sample of audio data (channel 0) synchronized across multiple sensors. Every 400 audio samples long audio data chunk can be placed in the right place on a global timeline by decoding the serialized timestamps embedded in the dedicated channel at a rate of 120 Hz.

**Figure 8 sensors-23-03710-f008:**
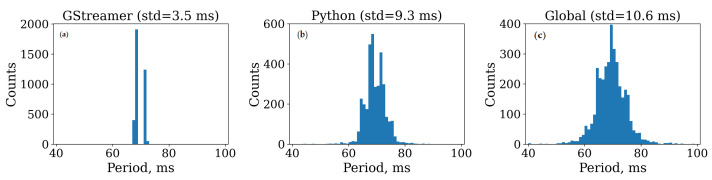
Each frame acquired by the cameras is timestamped three times: (**a**) by the camera driver (GStreamer), (**b**) by the REIP framework (Python), and (**c**) by the microcontroller receiving global timestamps from the master radio (Global). This figure illustrates a progressive degradation of timestamp quality, in terms of jitter, accumulated throughout the data acquisition pipeline.

**Figure 9 sensors-23-03710-f009:**
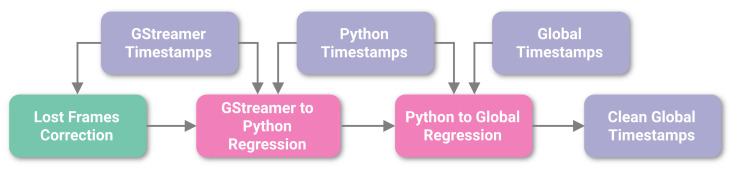
Diagram illustrating timestamp processing. We start with the least jittery GStreamer timestamps and identify any lost frames so that we can reconstruct the original timeline and average period for the saved frames. We then convert these reference timestamps into a global timeline through a series of regression steps that incorporate the information from other kinds of timestamps without adding jitter.

**Figure 10 sensors-23-03710-f010:**
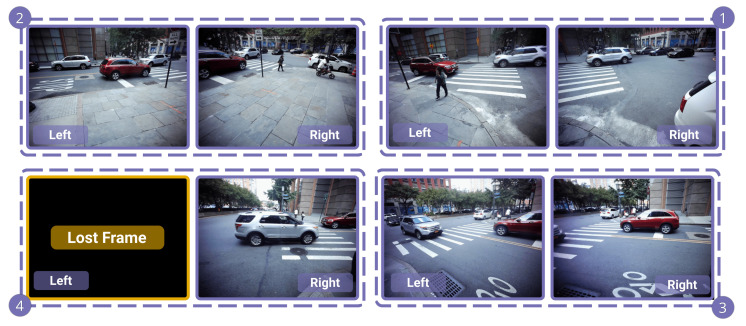
Mosaic rendering of the synchronized frames from recording session one at the Chase Center intersection that can be played as a video. Four sensors with two cameras each (numbered in the corners) provide eight different views for comprehensive analysis of the intersection. If a camera did not successfully record during a particular frame, its block is turned black, such as the left camera of sensor 4 in this example.

**Figure 11 sensors-23-03710-f011:**
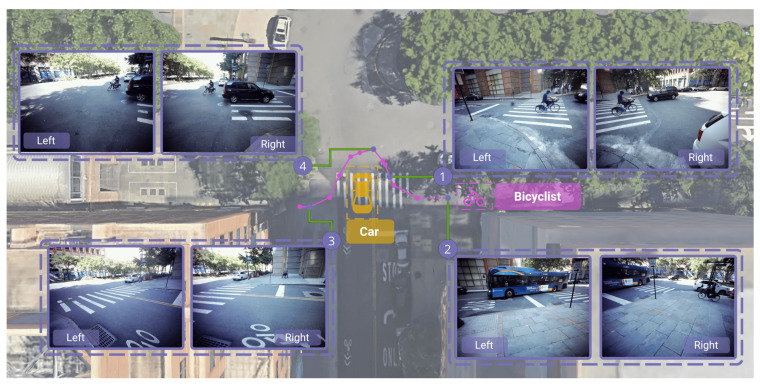
Audio-based localization of a bicyclist crossing the street at Chase Center and ringing the bell repeatedly (magenta points). In chronological order: Sensor 2 can see the bicyclist approaching the intersection, but localization of the bell ring is not possible because two sensors were occluded by a noisy bus; Sensor 1 view confirms the position of the bicyclist taking a right turn; Sensor 4 footage reveals the reason for the bicyclist’s curved trajectory—the black car did not stop to yield the right of way; Eventually, the bicyclist is no longer in the field of view of Sensor 3, but can still be localized thanks to the diffraction of the bell’s sound waves.

**Figure 12 sensors-23-03710-f012:**
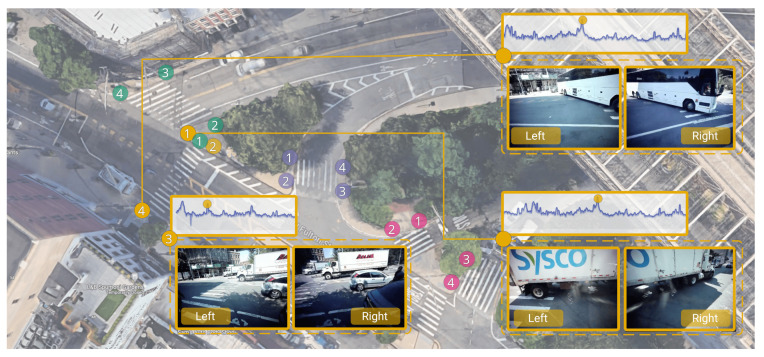
Sensor positions during data acquisition at DUMBO, Brooklyn. Colors indicate recording sessions and numbers denote the sensor. Above each synchronized video frame, we highlight the relative data point in the audio time series (in decibels). It can be shown that events that can be seen in the video, such as the passing of a bus, have a corresponding peak in the audio data.

**Figure 13 sensors-23-03710-f013:**
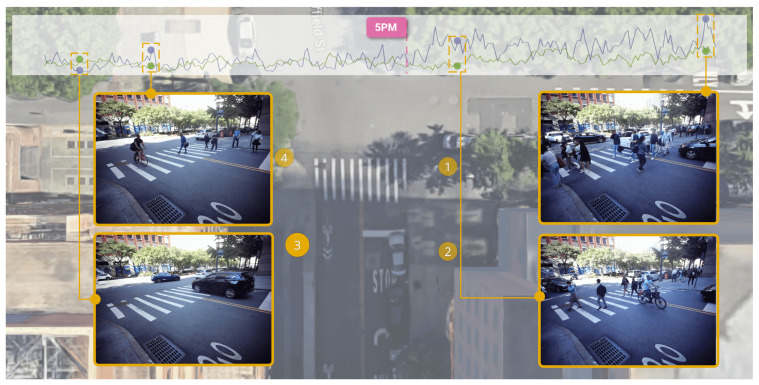
Chase Center intersection occupancy by object type during a recording session in the afternoon, with purple lines representing people and green lines representing cars on the top chart. In the figure, the four sensors collecting data during this session are represented by circles. There is a significant (≈3×) increase in the pedestrian count (blue) around 5 p.m. as people leave work. Moreover, it is possible to detect traffic light cycles based on the ratio of the number of pedestrians versus vehicle counts.

**Figure 14 sensors-23-03710-f014:**
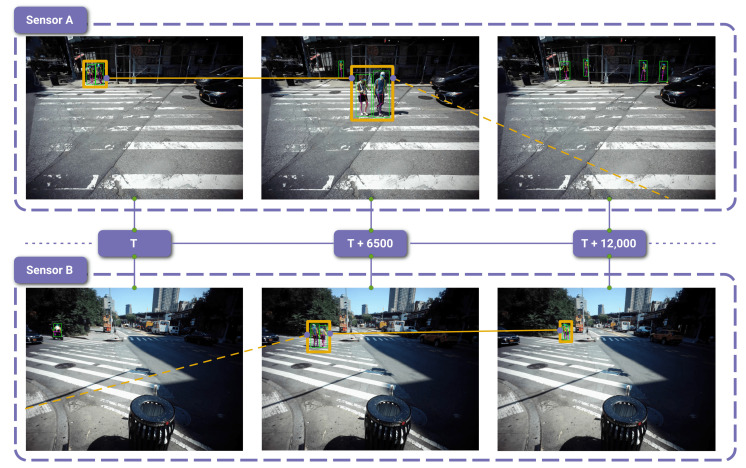
Two camera views from the session 2 recording at DUMBO. Camera A points northwest and Camera B points southwest. At the time T pedestrians (surrounded by an orange box) are visible in camera A but not in camera B. At the time T + 6500, as the pedestrians cross the street, they are observable by both cameras. By T + 12,000, the pedestrians are no longer observable in camera A but are still visible in Camera B as they continue to walk down the sidewalk. Time T represents the global timestamp at the moment the pedestrians begin crossing the street. By extension, T + 12,000 is the time 10 s later because the global timestamps are updated at a rate of 1200 Hz. This figure also highlights the advantage of high-resolution video. With objects at a farther distance from the camera, it becomes more challenging to detect them and estimate their poses. Higher resolutions help mitigate the information loss associated with more distant objects occupying a smaller portion of an image.

**Table 1 sensors-23-03710-t001:** Summary of available street-level datasets with their locations, sizes, descriptions, and whether or not they contain annotations.

Dataset	Location	Size	Description	Annotations?
Google Street View [6]	>100 countries	>220 B	Vehicle-mounted camera images; download not free	No
Mapillary Street-Level Sequences [7]	30 cities on 6 continents	>1.6 M	Vehicle-mounted camera images; condition-diverse; GPS-logged	No
Urban Mosaic [8]	New York	7.7 M	Vehicle-mounted camera images	No
SONYC [9]	New York	150 M	10-s audio samples	Yes
Urbansas [10]	European cities and Uruguay	15 h	10-s audio & video samples	Yes
KITTI [11]	Germany	1 k	Vehicle-mounted camera images; laser scans; GPS-logged	Yes
NuScenes [12]	Boston, MA	1.4 M	Vehicle-mounted camera images; radar & LiDAR; multi-camera	Yes
Waymo Open Dataset [13]	California & Arizona	1 M	Vehicle-mounted camera images; LiDAR; condition-diverse	Yes
Infrastructure to Vehicle Multi-View Pedestrian Detection Database (I2V-MVPD) [14]	Tunisia	9.48 k	Vehicle-mounted & stationary synchronized images	Yes
EuroCity Persons [15]	31 cities in 12 European countries	47 k	Vehicle-mounted camera images; condition-diverse; pedestrian-oriented	Yes
Pedestrian Intention Estimation (PIE) [16]	Toronto	911 k	Vehicle-mounted camera images; pedestrian & vehicle-oriented	Yes
KrishnaCam [17]	Pittsburgh, PA	7.6 M	Images from Google Glasses on pedestrian	No
Multi-view Extended Video with Activities (MEVA) [18]	Facility in Indiana, USA	9.3 kh	Stationary RGBIR & UAV video	Yes
Neovision2 Tower [19]	Hoover Tower at Stanford University	20 k	Stationary camera images	Yes
Cityscapes [20]	50 cities, most in Germany	25 k	Vehicle-mounted camera images	Yes
NightOwls [21]	Germany, Netherlands, & UK	279 k	Vehicle-mounted camera images at night	Yes
Cerema [22]	Controlled testing environment	62 k	Stationary camera images of pedestrians; varied rain/fog/light conditions	Yes
StreetAware	Brooklyn, NY	7.75 h	Stationary audio & video; synchronized, multi-perspective	No

**Table 2 sensors-23-03710-t002:** REIP sensor specifications including its two cameras, 12-channel microphone array, and NVIDIA Jetson Nano as a computing platform.

Feature	Specification
Internal Storage	250 GB
Power capacity	300 Wh
Camera resolution	5 MP
Camera field-of-view	160${}^{\circ }$ (85${}^{\circ }$ max per camera)
Camera frame rate	15 fps (nominal)
Audio channels	12 (4 × 3 array)
Audio sampling rate	48 kHz
NVIDIA Jetson Nano GPU and CPU cores	128 and 4
NVIDIA Jetson Nano CPU processor speed	1.43 GHz
NVIDIA Jetson Nano RAM	4 GB LPDDR4

**Table 3 sensors-23-03710-t003:** Dataset specifications after processing, featuring 3 data modalities (audio, video, and LiDAR) with synchronized footage.

Feature	Specification
Number of geographic locations	3
Number of recording sessions	11
Typical recording length	30–45 min
Total unique footage time	465 min (7.75 h)
Total number of image frames	≈403,000
Video resolution	2592 × 1944 pixels
Number of data modalities	3
Synchronized and anonymized	True
Video synchronization tolerance	2 frames
Audio synchronization tolerance	1 sample
Total audio & video size	236 GB
Total LiDAR size	291 GB
Total size	527 GB

## Data Availability

The data presented in this study are openly available in NYU Library at https://doi.org/10.58153/q1byv-qc065 (updated on 16 March 2023). The code for how the data were processed is located in a GitHub repository at https://github.com/reip-project/street-aware.git (updated on 16 March 2023).

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
