# Peer review of "StreetAware: A High-Resolution Synchronized Multimodal Urban Scene Dataset"

_sensors, 2023, doi:10.3390/s23073710_

Round 1

Reviewer 1 Report

The manuscript  present a high-resolution audio, video, and LiDAR dataset  of three urban intersections in Brooklyn, New York, totaling approximately 7 unique hours. The data  is collected with custom Reconfigurable Environmental Intelligence Platform (REIP) sensors that are designed with the ability to accurately synchronize multiple video and audio inputs. The resulting data is inclusively multimodal, mutliangular, high-resolution, and synchronized.  Using four ways the data can be utilized – (1) to discover and locate occluded objects using multiple sensors and modalities, (2) to associate audio events with their respective visual representations using both video and audio modes, (3) to track the amount of each type of object in a scene over time, and (4) to measure pedestrian speed using multiple synchronized camera views. In addition to these use cases, our data has been made available for other researchers to carry out analysis related to applying machine learning to understanding the urban environment (in which existing datasets may be inadequate) such as pedestrian-vehicle interaction modeling and pedestrian attribute recognition. Such analysis can help inform decisions made in the context of urban sensing and smart cities, including accessibility-aware urban design and Vision Zero initiatives.

Comments:

1.   The authors should check the affiliation please (1 New York University 2 NEC Laboratories America, Inc.)

2.   The authors use many keywords "Keywords: urban sensing; computer vision; urban multimedia data; urban computing; urban intelligence; smart cities; street-level imagery; data synchronization; object detection; object tracking". Please used just important keywords related with the work.

3.   The authors give details about figure one "Figure 1. Illustration of the basic concept of combining multimodal sensors at critical nodes (e.g., intersections) with on-device and in-vehicle computing capabilities to provide greater awareness to urban traffic participants. In this example, pedestrians and (semi)autonomous cars are sensed by sight (cameras) and sound (microphones) at intersections, and information is relayed to each car’s self-driving system. In this process, edge computing and via the cloud helps extract, in real-time, useful information from the data" in the introduction section.

4.   The authors should revise the caption of figure one "Figure 1. Illustration of the basic concept of combining multimodal sensors at critical nodes (e.g., intersections) with on-device and in-vehicle computing capabilities to provide greater awareness to urban traffic participants. In this example, pedestrians and (semi)autonomous cars are sensed by sight (cameras) and sound (microphones) at intersections, and information is relayed to each car’s self-driving system. In this process, edge computing and via the cloud helps extract, in real-time, useful information from the data" by reduce the caption of figure one.

5.   The authors give details about figure two "Figure 2. A photo of the REIP sensor in its protective metal housing ready for deployment (left) and its internal architecture (right). The sensor includes two cameras and a microphone array. It also features a high-precision custom synchronization solution for both video and audio data. It is based on a 2.4 GHz radio module receiving common global timestamps from a master radio device." in the The StreetAware Dataset section.

6.   The authors should revise the caption of figure two "Figure 2. A photo of the REIP sensor in its protective metal housing ready for deployment (left) and its internal architecture (right). The sensor includes two cameras and a microphone array. It also features a high-precision custom synchronization solution for both video and audio data. It is based on a 2.4 GHz radio module receiving common global timestamps from a master radio device." by reduce the caption of figure two.

7.   The authors should make table in the end of "Related works" section, the table should has the reference number, method, technical used, advantage and limitations.

8.   The authors should make flowchart for the purpose model.

9.   The authors should make table for evaluation parameters used in the simulation.

10.     The authors should separate section five in two section as 5. Discussion ;6. Conclusions.

11.     What are the limitations of proposed method? Please answer the question in the article.

12.     Please  follow the journal template.

Reviewer 2 Report

The authors make available a multimodal dataset from various sensors, for experimentation and analysis in urban environments.

Among the uses that the authors envision are to carry out unique applications of machine learning to urban street-level data, such as pedestrian-vehicle interaction modeling and pedestrian attribute recognition.

I believe that the way to acquire the data is well described and explained for each of the sensors, as well as the REIP specifications.

The data is accessible, I tried to access the data.

However, there are videos (as in DUMBO_3, sensor_1) where the camera is obscured by a tree, and I wonder if the fact that you can see a tree moving in most parts, could be useful, maybe for some optical flow work, but the authors should take care of the greater visibility.

My main concern of this data is the identification of people. There are countries that take care of the personal data of their citizens, data that can identify people. In this work, images of pedestrians are acquired without their consent, so the authors do not claim to have any permission. Although it may not be so much the definition to acquire the identity of the pedestrians, I believe that the authors should do some processing to blur the faces of the people to avoid their identification.

The authors state that video surveillance cameras do not provide sound or audio due to privacy, and I wonder if people's privacy would not be being violated.

In conclusion, I believe that this dataset from various sources can be very useful, however the authors should have a section pronouncing this situation.
